# Foraging Dynamics and Entropy Production in a Simulated Proto-Cell

**DOI:** 10.3390/e24121793

**Published:** 2022-12-08

**Authors:** Benjamin De Bari, Dilip K. Kondepudi, James A. Dixon

**Affiliations:** 1Department of Psychology, Lehigh University, Bethlehem, PA 18015, USA; 2Center for the Ecological Study of Perception and Action, University of Connecticut, Storrs, CT 06269, USA; 3Department of Chemistry, Wake Forest University, Winston-Salem, NC 27109, USA; 4Department of Psychological Sciences, University of Connecticut, Storrs, CT 06269, USA

**Keywords:** dissipative structure, foraging, entropy, maximum entropy production, entropy production, self organization, nonlinear dynamics

## Abstract

All organisms depend on a supply of energetic resources to power behavior and the irreversible entropy-producing processes that sustain them. Dissipative structure theory has often been a source of inspiration for better understanding the thermodynamics of biology, yet real organisms are inordinately more complex than most laboratory systems. Here we report on a simulated chemical dissipative structure that operates as a proto cell. The simulated swimmer moves through a 1D environment collecting resources that drive a nonlinear reaction network interior to the swimmer. The model minimally represents properties of a simple organism including rudimentary foraging and chemotaxis and an analog of a metabolism in the nonlinear reaction network. We evaluated how dynamical stability of the foraging dynamics (i.e., swimming and chemotaxis) relates to the rate of entropy production. Results suggested a relationship between dynamical steady states and entropy production that was tuned by the relative coordination of foraging and metabolic processes. Results include evidence in support of and contradicting one formulation of a maximum entropy production principle. We discuss the status of this principle and its relevance to biology.

## 1. Introduction

Arthur Iberall offered a humorously reductive yet apt characterization of biological behavior, suggesting that all organisms merely move to eat and eat to move [1]. While there is a staggering variety to the types of behaviors and functions exhibited by living systems, one trait common to all organisms is their dependence on a flux of energetic resources. These resources power irreversible entropy-producing processes within organisms that maintain their structure and function. Moreover, organisms manage those energetic resources through foraging behaviors and metabolic processes that modulate their concentration. These resources include sunlight for photosynthetic bacteria and plants, or organic materials for most other living creatures. All organisms actively collect resources, whether by configurational changes, as in the growth and tropism of rooted plants, or by directed translational motion through the environment, as in the taxis of most organisms. The intake, storage, and utilization of energetic resources constitutes perhaps the most fundamental behavioral repertoire of a living system. Herein we refer to this repertoire as *foraging dynamics*.

From a thermodynamic perspective, living systems can be understood as special types of dissipative structures, self-organized non-equilibrium systems driven by flows of energy and matter [2,3,4,5]. For example, the rhythmic activities of neurons, muscles, and metabolic processes has been explained as emergent non-equilibrium limit cycles [6,7], special kinds of *chemical clocks*. Dissipative structures are sustained by irreversible entropy-producing processes. Some non-living dissipative structures demonstrate a capability to *actively maintain* those irreversible processes that contribute to their stability. For example, we have reported on a non-living electrical dissipative structure (EDS) that tends to move in a way that ensures a supply of energetic resources (electrical charges) that maintains its stability [8,9,10,11], analogously to the foraging dynamics of living organisms.

Such systems must be studied with the tools of modern *non-equilibrium thermodynamics* [3], not traditional equilibrium thermodynamic principles. A common approach to non-equilibrium thermodynamics, including the study of dissipative structures, is to explain the time-evolution of such systems by a variational principle optimizing a physical quantity, especially the rate of entropy production (REP) [12,13], where REP is the first time-derivative of the quantity of entropy produced by a system. It has been proposed [14,15,16,17,18,19,20] that these same variational principles might apply to the time-evolution of organisms as well. For example, the REP has been measured in microbial colonies and used to explain their chemotaxis [20] and collective role in an ecosystem [17], the activation of muscle fibers [18,19] and even the evolution of the biosphere and perception-action [14].

One challenge in applying thermodynamic principles to biology is the sheer complexity of living systems. While a gross picture of the energy throughput that sustains organisms seems quite transparent, it depends on the coordination of many processes occurring at many different spatiotemporal scales, spanning from the cellular production and utilization of ATP to the organism-environment interaction in foraging. In non-living systems, the insights of thermodynamic principles have depended on defining clear boundaries between system and environment, and by focusing on distinct processes on either side of that boundary. This kind of complexity can be overcome, for example, Paltridge’s [21] seminal work evaluating the entropy production of Earth’s ocean-atmosphere system required partitioning the world into bounded areas in which singular thermal transport processes were occurring. The model yielded remarkable predictability of global weather patterns. In organisms, however, it is less obvious which constituent processes to choose or what partitioning to implement. Some [18,19] have investigated entropy production from the ATP flow in muscle-activity, for example, though this represents only one among many interacting entropy-producing processes.

To clarify the problem, let us further consider the study of entropy production in dissipative structures and how it might apply to biology. One proposed variational principle is the Maximum Entropy Production Principle (MEPP) which predicts that some far-from-equilibrium systems will tend to evolve to states that maximize the REP for that system. One specific formulation of a MEPP is that within a system with multiple dynamical modes, the most stable mode is also the mode with the highest rate of entropy production. Evidence for such a MEPP has been observed explain chemical, [22,23] electrical, [9,24], and fluid-mechanical [15] dissipative structures. Living systems are also well-suited for dynamical analysis, regularly exhibiting multi-stability, bifurcations, and limit cycles [6,7,25,26,27]. This is especially true in the context of physiological coordination, such as the joint oscillation of multiple limbs in locomotion [26,27,28,29]. When appendages (e.g., the pointer fingers on each hand) are used as simple oscillators, they exhibit multiple stable dynamical states (in-phase and anti-phase oscillation), as well as frequency-dependent transitions between those states and bifurcations in the relative stability of each mode [26,28].

Interlimb joint-oscillation, such as finger-wagging, is thus a reasonable candidate for testing MEPP in biological behavior; if the in-phase mode is more stable than the anti-phase mode, it should also yield a higher REP. The challenge appears when we consider measuring the entropy production of these states. It is methodologically tractable to measure the entropy production of humans, as has been done through calorimetry methods [30]. It is possible that by measuring the total heat produced one might be able to measure differences between in-phase and anti-phase dynamics. But these activities would only make very small contributions to the individual’s total entropy production. There are at any given time many other entropy-producing processes happening within an individual organism. It is also likely that these processes are often interdependent, such that changes in one have consequences for others. From this discussion it is clear that biological systems are (i) multi-scale, (ii) composed of multiple entropy-producing processes and (iii) the different processes are interactive. The REP measured includes contributions from all these processes, rendering it difficult to disentangle signal (i.e., the REP from any single behavior) and noise.

Despite this complexity, we envisage that thermodynamic quantities, including the REP, may still be revealing for biological behavior. As a preliminary way of addressing this, we study the entropy production in a minimally complex simulated dissipative structure, one which has a multiplicity of entropy-producing processes and dynamical states. The system is a kind of proto cell which has chemical reactions analogous to a metabolism and displays behaviors analogous to biological foraging. The metabolic network and the foraging dynamics can each be considered as dissipative structures that are coupled to one another. This system minimally instantiates the kind of complexity that creates challenges for the study of entropy production in biology. We tested the MEPP by measuring how the stability of the foraging dynamics relates to the system’s REP in different dynamical modes.

Our motivation for studying MEPP has been, in part, to better understand its domain of validity. Several authors have noted that MEPP does not apply to some systems [31,32,33,34,35,36]. For this and other reasons, the domain of validity of MEPP has been unclear, and there has been discussion on even the proper statement of MEPP. These issues have been addressed in many articles, notably by Martyushev and Seleznev [12,13,37,38]. As recently as 2014, Martyushev and Seleznev published an article titled “Restrictions of the maximum entropy production principle” [12], in which they note that MEPP applies to complex but not compound systems. As defined a compound system is one in which the system’s total entropy production is an additive function of sub-processes (e.g., chemical reactions, thermal diffusion) and is complex when this function is non-additive (i.e., multiplicative). Though most chemical systems are compound systems, there are chemical systems for which MEPP does apply [22,23]. This means that either some chemical systems are complex, or some compound systems can abide by MEPP. Thus, a motivation for this paper, which investigates the entropy production of a chemical dissipative structure, is to home in on the proper domain of chemical systems in which MEPP holds.

### 1.1. The Autonomous Forager Model

The present model is intended to simulate a complex dissipative structure. As a minimal analogue to biology, the model represents a protocell, a cell-like system with a boundary, internal chemical processes (metabolism), self-motion (locomotion), and a sensitivity to metabolizable compounds in its environment. The model stems from experimental work on synthetic systems called droplet swimmers, fluid droplets of oil or aqueous solution that move through their environments driven by some dissipative process [39,40,41,42,43]. The core features of the model are the capacity to forage for energetic resources, as well as the presence of a dynamic interior that can modulate the system’s interaction with the environment. To accomplish this, we simulate a system based on results from Suematsu et al. [42,43] which consists of an aqueous droplet swimmer with an embedded oscillating chemical reaction. Suematsu et al. [42,43] experimentally investigated the motion of this droplet, finding that its velocity oscillated due to the internal Belousov-Zhabotinsky (BZ) reaction. Additionally, we added properties present in other chemical droplet systems, including chemotaxis [39,40,41,44,45] and the exchange of materials with the embedding milieu [46].

It has been suggested that a promising step in artificial life is the incorporation of nonlinear chemistry to enable more life-like behaviors [47]. In line with this, Suematsu et al. [42,43] have developed a chemical swimmer with an embedded oscillatory reaction. The swimmer consists of a droplet of aqueous BZ reaction solution embedded in an oil bath. The aqueous droplet maintains a bounded structure due to the hydrophobic environment, and an aqueous BZ reaction occurs within the droplet creating oscillations in the concentration of intermediate compounds. On the surface of the droplet, an interfacial reaction between Br_2_, produced within the droplet, and Monoolein (MO), adsorbed from the exterior, modulates the interfacial tension. Interfacial tension drives a convective flow on the surface that propels the droplet (Figure 1). The internal concentration of Br_2_ oscillates periodically, with high concentration in the oxidized (blue) state and low concentration in the reduced (red) state. The behavior of the swimmer is thus directly tied to the nonlinear reaction occurring within it, moving faster or slower depending on the oscillating concentration of Br_2_, resulting in periodic increases in the droplet velocity [42,43]. Suematsu et al. [42,43] developed a computational model describing the oscillatory reaction and the droplet’s velocity that very accurately captures the empirical results. We incorporate and extend this computational model below.

#### 1.1.1. Model Schematic

The model herein is designed to represent a BZ droplet swimmer in a long rectangular oil-bath (Figure 2). The swimmer’s motion is largely constrained to the length of the bath, and so we simulate the droplet’s dynamics solely along this dimension *r*. A driving reactant for the BZ reaction, compound *A* (yellow), is supplied to the system from a central probe, and diffuses throughout the oil bath. This compound diffuses into the droplet according to the relative difference in concentration of *A* between the interior and exterior (Equation (14)). Concentrations of all compounds within the droplet are assumed to be spatially uniform due to convective mixing and diffusion. As the BZ reaction proceeds, it produces waste compound *Q* (orange) that diffuses from the droplet swimmer into the bath. *A* is an energetic resource, analogous to a food source, that drives the internal processes and behavior of the proto-cell, and *Q* is a low-energy product of the system (specifically, a product of the interfacial reaction that drives motion) analogous to organic waste. The concentrations of *A* and *Q* vary along *r* as a function of the supply of *A* from the source and the location of the swimmer. We include diffusion of compound *A* into the cell, as well as chemotaxis up increasing external spatial gradients of *A* (Equation (12)). The simulated system swims through its environment collecting fuel resources to sustain its structure and dynamics and is thus called the Autonomous Forager Model (AFM). 

#### 1.1.2. Oscillatory Reaction Dynamics

The BZ reaction has been studied extensively. It is understood to have a complex reaction network of as many as 22 coupled elementary reaction steps [48] but is often represented by simpler reaction networks. Suematsu et al. [43] demonstrated many of the empirical phenomena using an extended Oregonator model [49] to simulate the essential properties of their BZR swimmer. This reaction network instantiates the dynamics of the system but is not wholly reflective of the specific chemistry of the BZR. We use the same reaction network: (1)$$A+X+1.5Y\;+S\;↔Z+2P+Q^{\;}$$

R1 through R8 are the elementary reaction steps and (1) is the overall reaction. R1 to R5 are characteristic Oregonator reactions, while R6 to R8 are added to account for the interfacial reaction [19]. This network has eight chemical species (*A*, *P*, *Q*, *S*, *U*, *X*, *Y*, and *Z*) that represent some of the chemical species in the BZ swimmer. The most important species for understanding the present model are *A*: BrO_3_^−^, *U*: Br_2_, and *S*: MO. *A* is a primary driving reactant that reacts with compounds *X* and *Y*, present within the droplet, and is analogous to high-energy macromolecules or food of organisms (In a real BZR, compound *A* will not drive the reaction network alone but reacts with other compounds. For example, in some implementations of BZR, BrO_3_^−^ (compound *A*) reacts with Malonic acid to drive the reaction network. In this model these initiator reactants are assumed to be present within the droplet in excessive quantities. They are thus effectively constant and are not factored into the reaction kinetics.). *U* is an oscillating intermediate compound which Suematsu et al. [42,43] identified as a key factor in the motion of the droplet. *U* reacts with the diffuse *S* that adsorbs to the droplet surface from the surrounding oil, changing the interfacial tension on the surface of the droplet and driving the surface flow that propels the droplet through the oil. For simplicity, S is assumed to be concentrated in excess and is modeled as a constant. *Q*, the product of *U* and *S*, is analogous to organic waste and in this model diffuses out of the droplet into the environment.

Because we were interested in computing the REP for this system, we included the reverse reactions in addition to the forward reactions used by Suematsu et al. [43]. Additionally, in Suematsu et al.’s work, and in general practice, primary reactants like compound *A* are assumed to be in excess (i.e., being in large enough quantity that the reaction does not significantly change its concentration) and thus effectively constant. For this model, however, *all species except S are modeled as dynamic*. Unlike typical Oregonator models [49] here *A* is treated as a fully dynamic species, changing due to both the reactions and exchange with the exterior distribution. The full reaction scheme is provided in Table 1. The functions of key compounds in the network are listed in Table 2. Mass-action kinetics of the network are displayed in Table 3.

#### 1.1.3. Droplet Motion

Self-motion in non-living systems comes in a variety of forms (for reviews see; [47,50]), with most systems driven by chemical processes. These chemical “swimmers” often are at liquid-air interfaces or embedded in fluid milieus of water or oil. Motion in such systems typically arises from either interfacial tension gradients [50,51,52,53] or Marangoni flows [41,42,43,44] at the swimmer-environment interface. These interfacial tension gradients can derive from asymmetries in the aqueous distribution of compounds, as in [50,51,52] or asymmetries in the distribution of compounds on the swimmer interface, as in [41,42,43,44]. Suematsu et al. [42,43] identified a reaction between Br_2_ and MO on the surface of the droplet that modulates the interfacial tension, creating a convective flow along the surface that propels the droplet.

From the experimental derivation of Suematsu et al. [42,43] the velocity of the droplet is calculated according to:(10)$$v=v_{0}\sqrt{1-\frac{v_{1}^{*}}{v_{1}}}$$
where v is the droplet velocity, $v_{0}$ is the characteristic speed, $v_{1}$ is the chemo-mechanical coupling strength, and $v_{1}^{*}$ is calculated according to:(11)$$v_{1}^{*}=\frac{a+\frac{1}{\left[Br_{2}\right]}}{b\left(1-\beta *\left[Br_{2}\right]\right)}$$
where a, b, β are constants derived from the Navier-Stokes equations, and $\left[Br_{2}\right]$ is the concentration of the compound bromine [42,43]. All parameters in (11) are constants supplied to the model except $\left[Br_{2}\right]$ (*U* in the simulated reaction network) which varies as the BZ reaction proceeds. What is most important to notice is that the velocity of the droplet is a function of the internal concentration of $Br_{2}$ (*U*) and its motive dynamics will thus depend on the internal reaction dynamics. 

#### 1.1.4. Chemotaxis

Thus far, all aspects of the model as implemented are in accordance with published empirical and simulation work of Suematsu et al. [42,43]. To expand upon this work, we introduce a capacity for chemotaxis that will provide a new level of functionally rich behaviors. Chemotaxis occurs in similar oil-based droplet swimmers [39,40,41,44,45] thus it is plausible that a system like the AFM could be designed with such functionality.

For this model, we assume that the local concentration gradient of diffuse *A* gives direction to the droplet motion. To implement this, we simply calculate the gradient of the distribution local to the droplet and multiply it by the velocity equation:(12)$$v=C*\nabla A*v_{0}\sqrt{1\;–\;\frac{v_{1}^{*}}{v_{1}}}$$
where ∇A is the local concentration gradient of *A*. The gradient can be positive (concentration increases to the right) or negative (concentration increases to the left), giving the velocity a direction of either right (positive) or left (negative) in the one-dimensional space. The magnitude of the velocity is additionally scaled by the magnitude of ∇A. This relatively simple adjustment to the equations endows the swimmer with chemotactic properties, as the ∇A term will direct the droplet in the direction of increasing *A*. For convenience we assume this relationship is directly proportional to the gradient, though it need not be, and set coefficient *C* = 1.

The external distribution of compound *A* (Figure 3) varies as a function of the supply of particles to the system, and the absorption of particles by the droplet. Each discrete location in the space $x_{i}$ has a given value for the concentration of *A* which changes according to:(13)$$\frac{d\left[A_{i}\right]}{dt}=\gamma \left(A_{ss}\left(x_{i}\right)-[A_{i}\right])-J_{A}\left(x_{i}\right)+\delta \Gamma \left(x_{i}\right)$$
where γ is a scalar that sets the saturation rate for the supply term, $A_{ss}\left(x_{i}\right)$ is the steady-state concentration value for location $x_{i}$, $\left[A_{i}\right]$ is the concentration of compound *A* at location $x_{i}$, and $J_{A}\left(x_{i}\right)$ is the quantity of *A* absorbed by the droplet from location $x_{i}$. $\Gamma \left(x_{i}\right)$ is a Guassian-distributed random variable, scaled by δ, used for stochastic simulations. $J_{A}\left(x_{i}\right)$ is simulated according to an inverse-square rule (14). This exchange is constrained to be uni-directional, with *A* only flowing into the droplet.
(14)$$J_{A}\left(x_{i}\right)=\left\{\begin{array}{c} F_{A}\frac{\left[A_{i}\right]-[A]}{{\left(x_{i}-x_{c}\right)}^{2}+c},\;\;\;\;\;\;\;\left[A_{i}\right]-\left[A\right]>0\;\\ \;\;\;\;\;\;\;\;\;\;\;\;0,\;\;\;\;\;\;\;\;\;\;\;\;\;\left[A_{i}\right]-\left[A\right]<0 \end{array}\right.$$
where [*A*] is the concentration of *A* internal to the droplet. The environment was manipulated by changing the steady-state distribution of *A*. Twenty different distributions were used that varied only in their average baseline amount of compound *A* (i.e., the distribution was shifted vertically without changing shape). These environments were created by varying parameter *A_mod_* in (15) below:(15)$$A_{ss}\left(x_{i}\right)=A_{mod}+\frac{A_{height}}{\left(A_{width}*\left|x_{i}-A_{peak}\right|+c\;\right)}$$
where $A_{ss}\left(x_{i}\right)$ is the steady-state concentration at location $\left(x_{i}\right)$, $A_{mod}$ sets a baseline concentration value, $A_{height}$ scales the height of the distribution peak, $A_{width}$ scales the width and steepness of the peak, $A_{peak}$ is the location of maximum concentration, and *c* is a constant to prevent the denominator from going to zero. Unless specified in the text, values for all parameters used in the simulations can be found in the Appendix A.

### 1.2. Foraging Metrics

We have defined foraging dynamics as those behaviors and processes that contribute to the maintenance of energetic resources required for a dissipative structure’s continued existence. The “success” of foraging may be quantified by the amount of resources a system is able to take in. In the AFM, this can be calculated as the rate of *A* flowing into the droplet, $J_{A}$. Additionally, some research has supported that the *REP* is similarly a variable that correlates with a dissipative structure’s stability [2,9,10,11,15,22,23,24,54]. Research on chemotaxis in *Escherichia coli* found that the rate of consumption of resources is proportional to the *REP* [20], further motivating this comparison. Thus, both metrics $J_{A}$ and *REP* are reported to index foraging success.

$J_{A}$ is calculated by summing up the quantities of *A* (14) that flow into the droplet from all spaces $x_{i}$ (16).
(16)$$J_{A}=\underset{i}{\sum }J_{A}\left(x_{i}\right)\;\;\;$$

For an elementary reaction of the form aA+bB ↔cC+cD, the *REP*, σ, is calculated according to (17) through (21):(17)$$A_{n}=RT\ln\left(\frac{R_{fn}}{R_{rn}}\right)$$
(18)$$R_{fn}=k_{fn}{\left[A\right]}^{a}{\left[B\right]}^{b}$$
(19)$$R_{rn}=k_{rn}{\left[C\right]}^{c}{\left[D\right]}^{d}$$
(20)$$v_{n}=R_{fn}-R_{rn}$$
(21)$$\frac{d_{i}s}{dt}=\sigma =\sum_{n}\frac{A_{n}v_{n}}{T}=R\sum_{n}\ln\left(\frac{R_{fn}}{R_{rn}}\right)\left(R_{fn}-R_{rn}\right)\;\;\;\;\;\;\;\;\;\;\;\;\;n=1,\ldots 8$$
where $A_{n}$ is the reaction affinity, *R* is the molar gas constant, and $R_{fn}$ and $R_{rn}$ are the forward and reverse reaction rates respectively for each *n* reaction. $v_{n}$ is the net reaction rate of the *n*^th^ reaction, which is a function of the forward and reverse reaction rate constants $k_{fn}$ and $k_{rn}$, compound concentrations, and stoichiometric coefficients of the products and reactants in each of the *n* reactions. The system’s rate of entropy production per unit volume (where *s* is the entropy density), *σ*, is the sum of the product of each *n* reaction’s affinity and reaction velocity, divided by the temperature *T* [3]. Temperature *T* is assumed to be constant for all simulations and is ignored in the calculations. Entropy production is calculated only for the reactions within the droplet, not the exchange of materials with the exterior or any processes (e.g., diffusion) happening in the environment.

As a caveat, we note that this measure of the *REP* is only an estimate for the actual thermodynamics of a real system like the AFM. While the Oregonator reaction scheme reflects the dynamics of a real BZR, it neglects certain reaction steps and properties (i.e., chemical potential) of real compounds, and not all reactions have been reduced to their elementary steps. Additionally, the reaction rates are not necessarily reflective of the true values. Thus, the *REP* reported here does not necessarily reflect the thermodynamics of an actualized AFM or BZR system. Nevertheless, analysis focused on the relative values of the *REP* in different dynamical modes, and these relations may still hold in a physically realized system.

To evaluate the relation between stability and these foraging metrics, we conducted a rudimentary perturbation manipulation, comparing the foraging metrics between the stable and unstable modes across the dynamical regimes. Nonlinear dynamical systems can often exhibit a diversity of stable dynamical modes [55]. These modes can change as a control parameter is varied through a critical point, as in the transition to convection rolls in the Benard system [56]. In some systems, multiple modes can exist within a range of the control parameter, as in the relative-phase dynamics of bimanual coordination [28]. Many dissipative systems similarly exhibit these critical transitions and multi-stability. It has been observed in several systems that the “preferred” or stable mode is also the state with higher *REP*. This is true for the EDS described above [9,24] as well as for chemical systems [22,23] and fluid-mechanical systems [15,16]. Analysis of simulated bistable springs [57] and ensembles of chemically bonded particles [58] subject to periodic energy injection has revealed that the emergent stable structures can either maximize or minimize entropy production. Dynamical stability is often related to the optimization of the REP.

For the AFM, there are at least two consistently observed dynamical modes in the droplet’s motion, (i) a steady-state fixed point at the peak of the external distribution and (ii) a limit cycle with the droplet oscillating around the peak. Piloting revealed that $A_{mod}$ behaves as a control parameter, modulating the stability of the fixed-point or oscillatory modes. When the average external concentration is low (i.e., $A_{mod}$ is small), the droplet tends to exhibit fixed point dynamics, and as that baseline increases oscillations emerge and become the dominant dynamic. When the oscillatory mode emerges, the fixed-point steady state at the peak becomes an unstable fixed point (i.e., for deterministic simulations the system will remain at *x* = 0, but will not converge to the fixed point from arbitrarily small deviations in initial conditions). In this oscillatory region then there are two possible dynamics, a stable limit cycle and an unstable fixed-point. Consistent with the literature [9,16,22,23], we expected that the stable limit cycle should have higher *REP* compared to the unstable fixed point. We compared the *REP* and $J_{A}$ values between the different modes to evaluate this hypothesis. For all levels of $A_{mod}$, and all trials reported herein, the internal metabolic reaction displayed limit cycle dynamics. Only the motive dynamics ever exhibited fixed point behavior.

Two sets of simulations (Study 1) were conducted to understand the regions of stability for the different dynamical modes in the motion, and the relative *REP* and $J_{A}$ values between the different modes, across regions of stability. Within each level of $A_{mod}$ simulations were conducted with the droplet moving according to the velocity Equations (11) and (12), or with the droplet’s position fixed at the mid-point of the distribution where the fixed point is (by setting the initial position to *x* = 0 and parameter $v_{0}$ = 0). This second manipulation locks the droplet into the fixed-point dynamic, even if that fixed point is unstable. Thus, in the multi-stable regime we can compare the *REP* and $J_{A}$ between the stable self-selected limit cycle and unstable fixed-point dynamics. Together, these simulations provide a body of data for conducting a qualitative stability-analysis of the droplet’s behaviors jointly with an analysis of how the foraging metrics *REP* and $J_{A}$ relate to the stability and self-selection of dynamical modes.

As detailed below, the results from Study 1 were counter to expectations that the oscillatory mode should have a higher REP than the fixed-point mode; the opposite is true. Given this surprising result we performed several sets of simulations to explore how the REP changes dependent on the interaction between the droplet motion and the internal reaction network. The intake of fuel from the environment depends on the internal concentration of *A* (Equation (14)), and thus continually oscillates as the internal reaction proceeds. While the droplet is continually importing fuel, there are periods of both low in-flux and high in-flux. The droplet is also continually modifying the spatial distribution of *A*, creating regions of relatively high and low concentration throughout. The droplet is programmed to follow increasing gradients of *A*, but due to the periodic changes in $J_{A}$ it may not be importing *A* at the optimal times. That is, $J_{A}$ may be out of sync with the droplet’s position relative to the fuel-rich regions of the distribution. To test this, a set of experiments (Study 2) were conducted that varied the timing of $J_{A}$ relative to the droplet’s motion.

## 2. Materials and Methods

### 2.1. Study 1: Entropy Production and Dynamical Stability

This study was conducted to investigate how the foraging metrics (*REP* and $J_{A}$) from self-selected behavior compared to an induced fixed-point dynamic. As noted above, nonequilibrium self-organizing systems can exhibit multi-stability. In several of such systems, the mode which has greater average REP is also the more stable mode or the mode which the system tends to self-select for [9,15,22,23,24]. Piloting revealed that the fixed-point tended to be more stable for low values of $A_{mod}$, while oscillations tended to be stable for higher levels of $A_{mod}$.

Given that there are these two dynamical modes, we can evaluate how the foraging metrics vary between them. To do this, two types of simulations were conducted for each level of $A_{mod}$: (a) the droplet was free and unconstrained, (b) the droplet was fixed at *x* = 0 and did not move. This meant that for critical levels of $A_{mod}$, those for which the droplet tended to oscillate, we could compare the average *REP* and $J_{A}$ values between the self-selected dynamics and the induced-fixed point dynamic. Thus, we can address whether the foraging metrics are higher in the stable dynamical mode, testing the MEPP prediction that entropy production selects for dynamical stability. The induced fixed-point dynamic was instantiated by giving the droplet an initial position of *x =* 0 and setting term $v_{0}=0$ in Equation (12) so that all motion stopped. 

Simulations were conducted with the droplet being unconstrained or fixed at *x* = 0, crossed with twenty levels of $A_{mod}$ ranging from 0.1 to 2 in increments of 0.1. 20 stochastic trials were conducted at each level of $A_{mod}$ with noise term δ = 0.25. This led to a 2 × 20 × 20 design, yielding 800 simulations. The droplet had an initial position of *x* = 0.1 for unconstrained trials, and *x =* 0 for induced fixed point trials. We then took the difference of the average values for each foraging metric between the free and perturbed cases as an index of which dynamic led to greater foraging success. Averages were taken after the first 2500 timesteps to ignore transient dynamics. These relative values of the foraging metrics were then compared to the stability of each mode. The expectation, consistent with previous literature, was that the *REP* and $J_{A}$ should be higher in the oscillatory *if* that mode is also the more stable mode (i.e., the self-selected mode).

The droplet’s position, all internal concentration values, and flux of *A*, $J_{A}$, were recorded from the simulations. The *REP* was calculated from the internal concentration times and reaction rates after the simulation. The position data in the unconstrained condition were converted into amplitude data by computing the analytic signal via the Hilbert transform and subsequently taking the absolute value. Oscillation amplitude was averaged at the trial level. To compare the relative *REP* and $J_{A}$ between oscillating and static conditions we averaged the values within levels of $A_{mod}$ and took the difference of the means between the oscillatory and perturbed conditions. Positive values of the relative foraging metrics indicate that the self-selected dynamic in the unconstrained conditions produced greater foraging success than the fixed-point mode.
(22)$$∆\sigma =REP_{free}-REP_{fixed}$$
(23)$$∆J=J_{A_{free}}-J_{A_{fixed}}$$

### 2.2. Study 2: Relative Timing of Flux and Motion

Counter to expectations, the more stable oscillatory mode was not the mode with highest REP. The REP depends on the internal metabolic reactions that are driven by compound *A*. As shown in Equations (13) and (14), $J_{A}$ depends on (i) the internal concentration of *A* and (ii) the position of the droplet relative to the external distribution of *A*. If there is a high concentration of *A* internal to the droplet, the flux will be low compared to when the internal concentration is low. $J_{A}$ also depends on the concentration external to the droplet, and the diffusion is scaled by the inverse square of the distance from the droplet and the region of the distribution it is drawing from. Property (i) causes $J_{A}$ to oscillate as the internal concentration of *A* oscillates with the metabolic reaction network. Property (ii) causes $J_{A}$ to be relatively high (low) if the concentration of *A* local to the droplet is high (low). 

Because of these two properties, it was hypothesized that the timing of the peak $J_{A}$ relative to the droplet’s position would cause changes in the average $J_{A}$ and consequently the average REP. For example, if $J_{A}$ peaks while the droplet is in a region of low external concentration, the flux will be less than if it peaked while the droplet is in a region of relatively high external concentration. Thus, the relationship between the internal oscillatory process and the behavioral oscillation should affect the REP. 

To manipulate this relationship, simulations were run with the droplet’s motion confined to a pre-established sinusoid and varying the phase-relation between the motive and metabolic oscillations. This allows $J_{A}$ to peak at different times relative to where the droplet is in its trajectory, and thus where it is relative to the distribution of *A*. For certain relative phase values then the average $J_{A}$ and entropy production should be higher due to the flux peaking nearing the fuel-rich regions of the distribution. The sinusoidal trajectory was calculated according to
(24)$$x\left(t\right)=\sin\left(ft+\varphi \right)\alpha \;$$
where *x* is the droplet’s position, *f* is the frequency, *t* is time, φ shifts the starting phase of the cycle and α is the oscillation amplitude.

The droplet’s oscillation amplitude (Figure 4) was calculated from stochastic trials in study 1 for each level of $A_{mod}$ from 0.8 to 2 (oscillations did not occur below $A_{mod}$ = 0.8). The droplet’s total motion was a symmetric cycle extending to either side of the midpoint of the distribution. Motive frequency was calculated for cycles defined by the droplet’s departure and return to the midpoint of the space, or one half of the total cycle. As shown in Figure 5, the frequencies of the motion and metabolism calculated from study 1 were very close to each other across levels of $A_{mod}$. For the simulations in which the droplet’s motion was prescribed, amplitude was set to the same value as that estimated from the deterministic trials, and frequency was set to the value of the *metabolic cycle*, so that relative phase would be effectively constant throughout trials. To manipulate relative phase, a series of φ values that ranged from −6.35 to 6.35, in increments of 0.05, were used to ensure that trials with relative phase values between 0 and 2π were observed. The parameters for each level of $A_{mod}$ are displayed in Table 4.

Relative phase was calculated using Hilbert transform methods (e.g., [59]). Relative phase was always calculated as
(25)$$∆\Phi =\Phi_{x}-\Phi_{J}\;$$
where $\Phi_{x}$ is the unwrapped phase of the position cycle and $\Phi_{J}$ is the unwrapped phase of the flux. The unwrapped phase was used because the frequencies of both processes were constant and identical so there was no phase-wandering driven by differences in frequency. If relative phase values were negative, they were corrected with a shift of positive 2π radians so that all values of ∆Φ ranged from 0 to 2π (leading or lagging is thus ignored).

## 3. Results

### 3.1. Study 1

The droplet demonstrates a clear bifurcation to the oscillatory mode as control parameter $A_{mod}$ increases (Figure 4). The critical value is $A_{mod}$ = 0.8, below which the fixed point is stable and above which the oscillatory mode emerges. 

Figure 5 displays the average oscillation frequency for the motion and metabolism from $A_{mod}$ = 0.8 to 2.0, where oscillation occurred. The two frequencies are consistently similar. 

As $A_{mod}$ increases, the system’s REP increases monotonically for both the unconstrained and fixed-point trials (Figure 6). To better differentiate between the two, the relative REP, ∆σ, calculated as the mean REP in fixed point trials subtracted from that of the unconstrained trials (Equation (22)), is plotted in Figure 7. Negative values indicate that the fixed-point dynamic has a higher REP. For all values of the control parameter for which the limit cycle is stable the fixed-point mode has a higher REP. A paired-samples *t*-test comparing the intra-trial average σ between oscillating and stationary motion revealed a significant difference (t(559) = −37.46, *p* = 7.28 × 10^−111^) between the entropy production of each mode of motion.

A similar relationship is observed for ∆J (Figure 8); the flux of *A* into the droplet is greater in the fixed-point dynamic than the limit-cycle in all except $A_{mod}$ = 1.7 and 1.9. A paired-samples t-test comparing the intra-trial average J between oscillating and stationary motion revealed a significant difference (t(559) = −18.69, *p* = 4.14 × 10^−51^) between the entropy production of each mode of motion. Above $A_{mod}$ = 0.7 the ∆J and ∆σ vary non-monotonically. This is due to the effect of the relative timing between the metabolic and motive dynamics, as investigated in Study 2.

### 3.2. Study 2

Figure 9 displays time series for the REP, $J_{A}$, and internal concentration of *A* from a subset of an oscillatory trial for $A_{mod}$ = 0.8. All values are normalized to appear on the same scale. $J_{A}$ is maximal when *A* is minimal due to the concentration-dependence in Equation (14). *A* oscillates due to the metabolic reaction network. The REP is maximal shortly after *A* is minimized since entropy is produced through the conversion of *A* into intermediate and waste compounds.

Figure 10 displays the average $J_{A}$ at each value of ∆Φ for $A_{mod}$ = 0.8. Different phase relations between the motion and the internal oscillations modulate the magnitude of $J_{A}$.

Figure 11 displays the average relative flux ∆J for each ∆Φ within all the environments. Across levels of $A_{mod}$, ∆J was maximal on average at ∆Φ = 3.92 (SD = 1.03). This bias towards ∆Φ greater than π radians is likely due to the asymmetry of the droplet’s trajectory. Recall that one cycle of the droplet’s motion is defined as departing from and returning to the midpoint of the external distribution (i.e., *x* = 0). If the flux is maximal in the first half of this cycle (∆Φ < π) then it peaks while the droplet is moving *away* from the fuel-rich midpoint, while if it peaks in the second half of the cycle (∆Φ > π) then the droplet is moving *towards* the midpoint. While the droplet is always following the largest gradient on either side, the magnitude of this gradient is larger if it is increasing towards the midpoint than away, due to the parabolic nature of the distribution.

Table 5 displays the maximum flux values within each level of $A_{mod}$ and the ∆Φ value at which they occur.

Figure 12 displays the average ∆σ at each value of ∆Φ for $A_{mod}$ = 0.8. Positive values of ∆σ indicate that the oscillatory dynamic produced more entropy, and negative values indicate that the fixed-point dynamic produced more entropy. There is a clear effect of ∆Φ on ∆σ_,_ and critically a large region of the relative phase in which ∆σ is negative and a small region in which ∆σ is positive. Thus, different phase relations between the motion and the intake of *A* modulate show which dynamical mode has a higher REP.

Figure 13 displays the average ∆σ for each ∆Φ within each level of $A_{mod}$. For all values of $A_{mod}$ except $A_{mod}$ = 1.9 there is a region of positive ∆σ, depicted in yellow. Across values of $A_{mod}$, ∆σ was maximal on average at a relative phase of ∆Φ = 4.21 (SD = 0.77). Table 6 details the maximal ∆σ value and the average relative phase at which it occurs.

Both ∆J and ∆σ varied with relative phase, though ∆σ was not maximized at the exact same values of relative phase as ∆J. While $J_{A}$ and σ are proportional, σ depends on many other parameters, especially the reaction rates. Looking within individual trials we can compare the relative phase of the REP and $J_{A}$ time-series. On average, across values of $A_{mod}$ and relative phase conditions, the REP had a phase lag behind $J_{A}$ of −0.71 radians (SD = 0.041). That is, the REP tended to peak after $J_{A}$, due to the intervening reaction steps converted *A* into intermediate compounds. This partially explains the similar lag of −0.29 in ∆Φ where ∆σ was maximal compared to $J_{A}$.

It was assumed that the variation in ∆J and ∆σ with relative phase is due to the droplet intaking quantities of *A* dependent on where it is in the distribution when $J_{A}$ is at its peak. To quantify properties of the distribution we looked at the average local concentration near the droplet ${\bar{\left[A\right]}}_{L}$ and the droplet’s displacement from the point of highest *A* concentration, $x_{disp}$. The mean local concentration ${\bar{\left[A\right]}}_{L}$ was calculated by sampling the distribution within 10 discrete locations on either side of the droplet and taking their average. $x_{disp}$ was calculated by finding the location of the space with maximal concentration at each timestep and finding the absolute magnitude of the distance between the droplet and that location.

We sampled each of these values only at the times that $J_{A}$ was maximal. A peak-picking function from the Python package SciPy [60] was used to find the points in the time series at which $J_{A}$ was maximal. From the timeseries of ${\bar{\left[A\right]}}_{L}$ and $x_{disp}$ we resampled the values at time-points where the flux was maximal, then took their average. Figure 13 reports the mean local concentration ${\bar{\left[A\right]}}_{L}$ at the time of maximal $J_{A}$ across relative phase values and values of $A_{mod}$. Figure 14 displays the ${\bar{\left[A\right]}}_{L}$ values for each value of relative phase and values of $A_{mod}$. Across all values of $A_{mod}$, ${\bar{\left[A\right]}}_{L}$ was maximal at mean relative phase ∆Φ = 4.73 (SD = 0.26) meaning there is on average more *A* near the droplet in the latter half of its cycle when it is moving towards the midpoint, consistent with the trends observed in both $J_{A}$ and ∆σ.

Figure 15 displays the droplet’s average displacement from the point of highest concentration of *A* across relative phase and all values of $A_{mod}$. Here the *minimum* of the displacement is most relevant, as minimizing the displacement from the peak means the droplet is nearer the region of higher concentration and thus would absorb higher quantities of *A*. Across all values of $A_{mod}$, $x_{disp}$ was minimized on average at relative phase ∆Φ = 4.44 (SD = 0.39). This is consistent with the data suggesting that the droplet absorbs the most *A* when $J_{A}$ is maximal in the second half of the cycle when approaching the midpoint. 

## 4. Discussion

In previous work, a foraging dissipative structure demonstrated an instability-driven transition to limit-cycle dynamics, and this new oscillatory mode had higher average REP than the unstable fixed-point mode [9]. We similarly observe a transition to oscillations in the AFM as the system is pushed further from equilibrium. Similar to the electrical dissipative structure, the emergence of oscillations in the AFM occurs due to a depletion-driven symmetry-breaking of an embedding distribution of energetic compounds. In the AFM, this depletion-process, $J_{A}$, is not constant (as it is in the electrical dissipative structure) but rather fluctuates as the internal metabolic reaction proceeds. The flux is effectively gated by the internal concentration of *A*, decreasing whenever *A* is highly concentrated within the droplet. 

The magnitude of the flux is also scaled by the external concentration local to the droplet, making $J_{A}$ larger if in a fuel-rich region and smaller if not. Since the droplet is continually moving through the distribution, and thus the local external concentration of *A* is varying, $J_{A}$ increases or decreases depending on where it peaks relative to the external distribution. ${\bar{\left[A\right]}}_{L}$ at the time of peak $J_{A}$ was maximal when the flux peaked in the second half of the cycle (i.e., when it was moving towards the midpoint at *x* = 0). The droplet’s displacement from the point of maximal [*A*] at the time of peak $J_{A}$ was similarly minimized when the flux peaked in the second half of the cycle. Together these data demonstrate how the relative timing of the flux and motive cycles affect the flux of the droplet and, consequently, its *REP*. 

This in turn affects ∆σ_,_ the difference in the *REP* between the oscillatory and fixed-point dynamics, as *A* drives the internal entropy-producing reactions, with lower $J_{A}$ leading to lower entropy production. ∆σ showed similar trends to $J_{A}$, tending to be maximal in the second half of the cycle. ∆σ did not directly reflect $J_{A}$ due to the nonlinearities mediating the in-flux of *A* and the production of entropy through the internal reactions. Both ∆σ and ∆J, displayed regions of positivity and negativity in nearly all levels of $A_{mod}$ dependent on the relative phase between the metabolic and motive cycles. These results were intended to test a MEPP prediction: the stable dynamic in a nonlinear system will also be the state with the highest entropy production. The results present a slightly more complicated story, with ∆Φ modulating the entropy production of the stable oscillatory mode, leading to some conditions consistent with the MEPP (i.e., positive ∆σ) and others inconsistent with it. The states with positive ∆σ were predominantly overlapping with states of positive ∆J, consistent with previous evidence that the REP is related to foraging success [20].

Several possibilities remain when considering these results. For example, it is possible that the proper formulation of the MEPP applies to stable states in a multi-stable system, rather than reflecting differences between stable and unstable dynamics. Some previous studies have found support for this MEPP by comparing coexisting stable modes. These results only compared a stable oscillatory dynamic to an unstable limit [9,15,22,23] cycle mode, similar to observations made in the electrical dissipative structure. Previous research looking at the entropy production of oscillatory reactions in autocatalytic chemical systems has produced a variety of results that show the limit cycle mode having higher [61] and lower [62] REP than the unstable fixed point dynamic. Additionally, simulation of bistable springs [57] and ensembles of particles with changeable chemical bonds [58] revealed that driven dissipative systems can settle into stable dynamics that either minimize or maximize the REP. These latter two papers suggest a common explanatory framework of dissipative adaptation, wherein irreversible transformations within a system lead to stable self-organized configurations that modulate the ability to absorb and dissipate energy, especially with structures that resonate to the time-structure of energy injection. It is possible that dissipative adaptation underlies the results herein, as the REP similarly depends on the relative synchronization of internal oscillations and the intake of fuel ***A***. The current modeling techniques rely on macroscopic mass-action approximations, but future research should investigate the statistical-mechanical counterparts to this.

This model depends on several fixed parameters, including the reaction rates, diffusion constants, and viscous damping, that are likely to be more variable in a physically realized version of the AFM or in an organism. While the model could artificially sample the entire parameter space, it is possible that in a physically realized analog of the AFM it would reside only in the regions consistent with or inconsistent with the MEPP. Relative phase in the self-selected dynamics does not vary with changes in initial conditions, but seems to emerge due to the depletion and relaxation of the embedding chemical distribution, all of which is directly dependent on fixed parameters. Subsequent work should investigate if there are clear selection criteria for certain relative phase values, as these would more directly address whether the dynamics are consistent with the MEPP. Additionally, we only investigated the entropy production of the internal reaction, not the transport or diffusion processes happening in the system. It is possible that including these in a subsequent analysis would more directly address the MEPP. This is a particularly promising avenue as the results herein show that the coordination of the motion and metabolism modulates the relative *REP*, ∆σ, and thus more directly accounting for the entropy produced by importing metabolizable compounds may clarify the phenomenon.

Lastly, it is possible that multiple variational principles may govern the time evolution of complex dissipative structures under different conditions, for example either minimizing or maximizing the REP. It is sometimes suggested that organisms ought to manage energetic resources efficiently and thus minimize their rate of entropy production. Different environmental pressures may modulate the degree to which efficiency is required, necessitating it when resources are scarce but not when plentiful. Some organisms like *Dictostelyium Discoiedium* display dramatic changes in behavior and morphology based on the availability of resources [56], though its entropy production in different modes has not been measured to our knowledge. 

Organisms are typically understood to behave economically in that actions cannot cost more energy than they provide to the system [63], and thus different behaviors are selected to optimize energetic efficiency [64]. Ideally, then, foraging efficiency requires maximizing energy consumption while minimizing energy utilization. Similarly, minimizing or maximizing the REP may reflect these different constraints on biological behavior, namely the tendency to minimize energy utilization while maximizing energy consumption, respectively. Herein the REP depends on the amount of energy imported, specifically flux $J_{A}$, and thus the stable dynamic corresponds with the maximum or minimum entropy production state dependent on whether it is has higher or lower $J_{A}$ relative to the unstable steady state. This system thus may represent only one of the constraints on biological foraging, namely maximizing energy consumption, without any process to minimize energy utilization. A complex dissipative structure like the AFM may be able to switch between variational principles depending on embedding circumstances, minimizing or maximizing entropy as it is adaptive to minimize energy utilization or maximize energy consumption, respectively.

## 5. Conclusions

The model developed herein is a first attempt to investigate dissipative structures with complexity that while modest compared to organisms, does have an internal metabolism-like reaction network coupled to chemotactic motion, moving to eat and eating to move as Iberall would say. Organisms have multiplicities of interactive processes that render their relation to macroscopic thermodynamic processes non-obvious. The AFM consists of an autocatalytic reaction within a droplet coupled to a flow of chemical reactants, leading to an interdependence between metabolic and motive dynamics. Results demonstrated that the nature of the interaction between motion and metabolism affects the entropy produced by the system, and whether the system is consistent with an MEPP or not. The AFM displays an interesting counterexample in which the more stable oscillatory dynamic does not have the higher REP, despite its similarities to another foraging dissipative structure. Given the current state of the art in the fields of chemistry, physics, and their intersection in active matter, we can expect to see that analogous systems will be developed in which predictions from a MEPP may be tested more directly. These results serve as a modest step towards conceptualizing thermodynamics of complex dissipative structures and their activities.

## Figures and Tables

**Figure 1 entropy-24-01793-f001:**
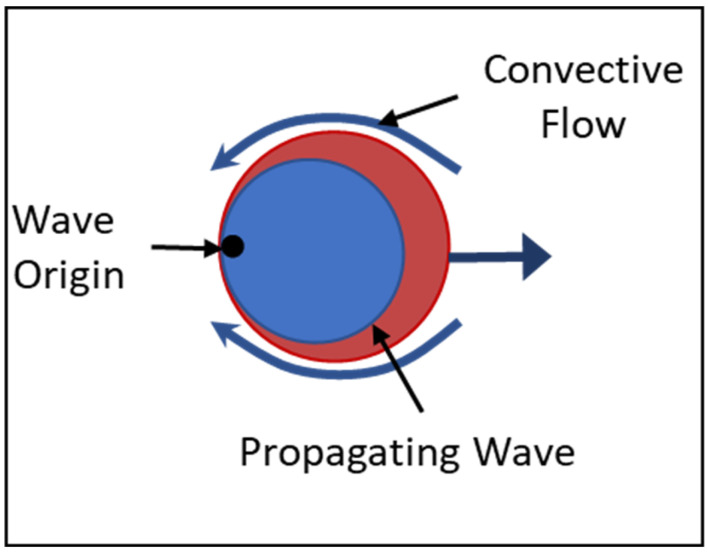
An interfacial reaction drives a convective flow on the surface of the droplet, propelling it through the oil. The origin of the oxidized state of the BZ reaction is asymmetric, propagating from one side of the droplet generating a chemical wave. The periodic waves modulate the interfacial reaction, driving oscillation in the droplet velocity.

**Figure 2 entropy-24-01793-f002:**
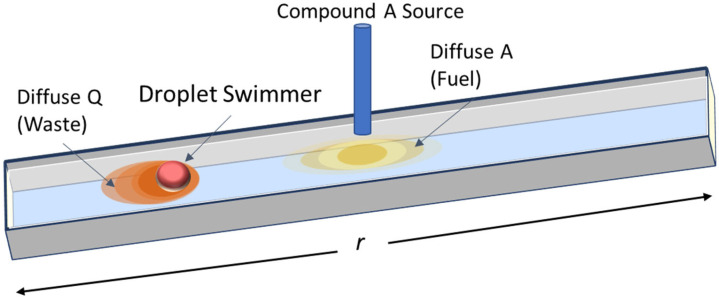
Schematic of the hypothetical system represented by the model. A long narrow oil bath restricts the swimmer’s motion to the length of the vessel. Compound *A* is continuously supplied to the system. Compound *Q* diffuses from the cell, driving its motion. Colors are not representative of the real compounds or their concentrations.

**Figure 3 entropy-24-01793-f003:**
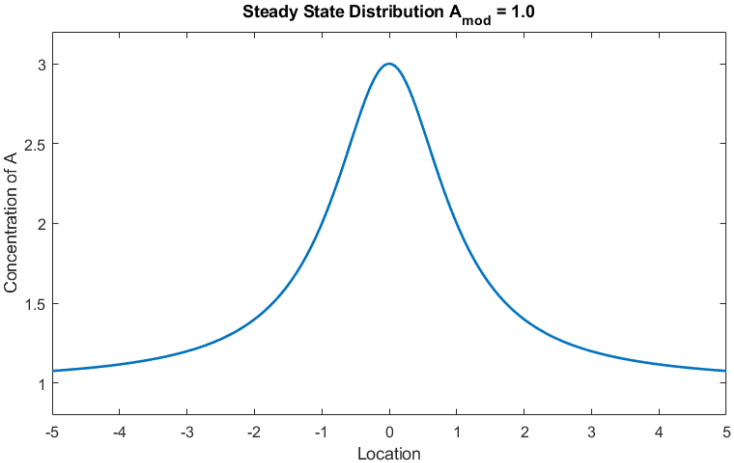
Steady state distribution of compound A for $A_{mod}$ = 1, $A_{height}$ = 1, $A_{peak}$ = 0, $A_{width}$ = 0.5, *c* = 0.5. Only parameter was $A_{mod}$ varied.

**Figure 4 entropy-24-01793-f004:**
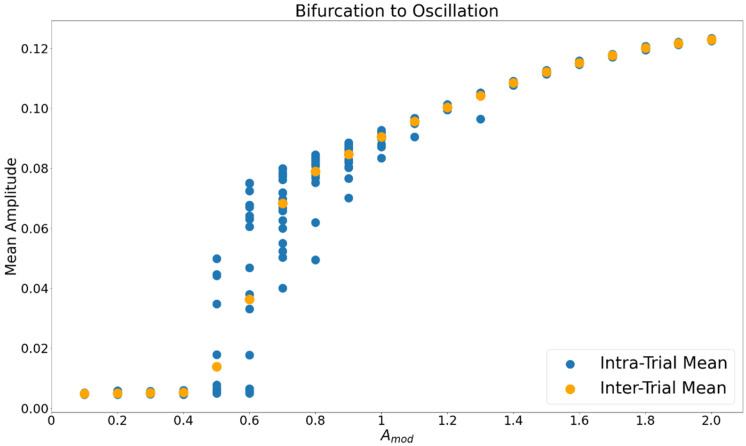
The average amplitude of the droplet’s motion within each level of $A_{mod}$ in the unconstrained trials. Blue markers are averages from individual stochastic trials, orange markers are the average across trials within level of $A_{mod}$. At $A_{mod}$ = 0.7 a bifurcation to the limit cycle occurs and oscillations dominate.

**Figure 5 entropy-24-01793-f005:**
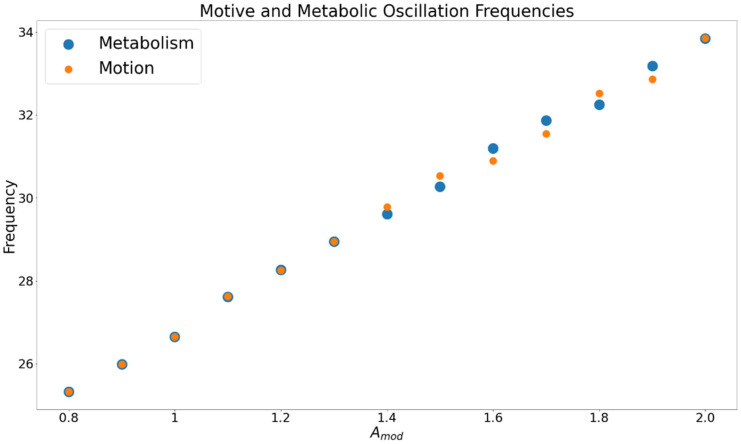
Frequency of both metabolic (blue) and motive (orange) oscillations across values of $A_{mod}$. $A_{mod}$ values below 0.8 are not displayed as the droplet did not display motive oscillations.

**Figure 6 entropy-24-01793-f006:**
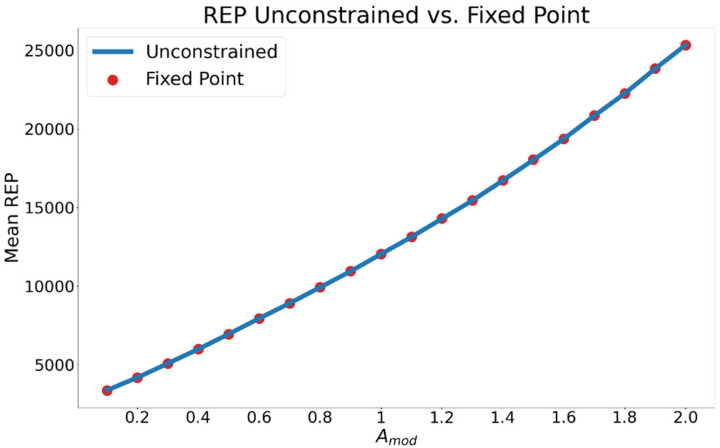
Average REP in the droplet for different levels of control parameter $A_{mod}$. REP increases with $A_{mod}$ due to the increase of the primary reactant A. Averages are calculated from all 20 stochastic trials.

**Figure 7 entropy-24-01793-f007:**
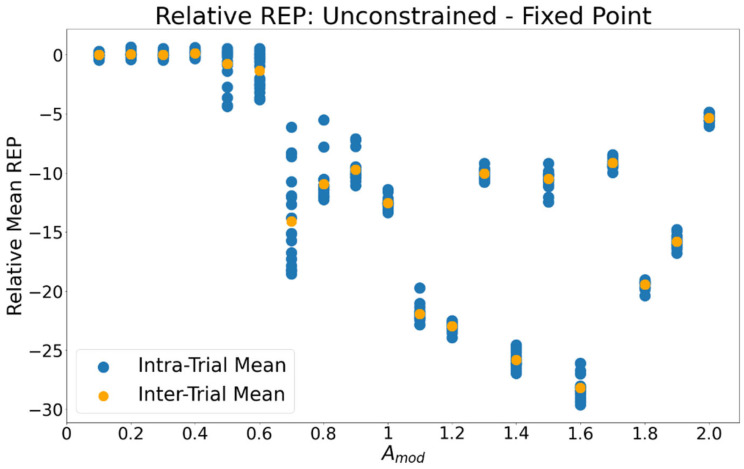
Relative REP, ∆σ_,_ between unconstrained and fixed-point dynamics. The REP of the induced fixed-point trial was subtracted from that of the unconstrained trial. Negative values indicate that the fixed-point dynamic had higher REP than the unconstrained dynamic. Blue markers indicate the average ∆σ within individual stochastic trials, orange markers indicate the average across all stochastic trials within levels of $A_{mod}$.

**Figure 8 entropy-24-01793-f008:**
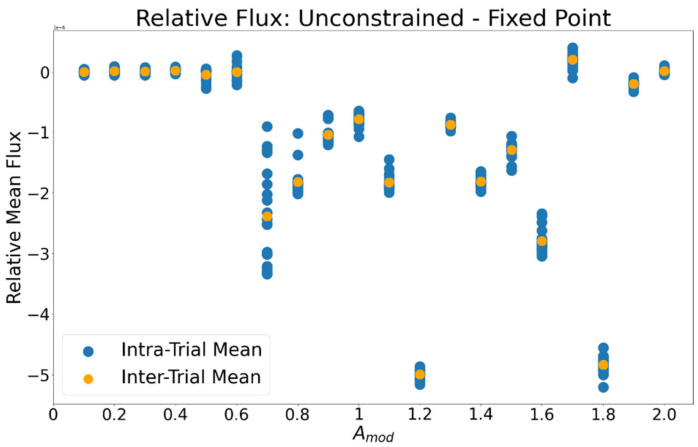
Relative mean flux ∆J between unconstrained and fixed-point trials. $J_{A}$ of the induced fixed-point trial was subtracted from that of the unconstrained trial. Negative values indicate that the fixed-point dynamic had higher $J_{A}$ than the unconstrained dynamic. Blue markers indicate the average ∆J within individual stochastic trials, orange markers indicate the average across all stochastic trials within levels of $A_{mod}$.

**Figure 9 entropy-24-01793-f009:**
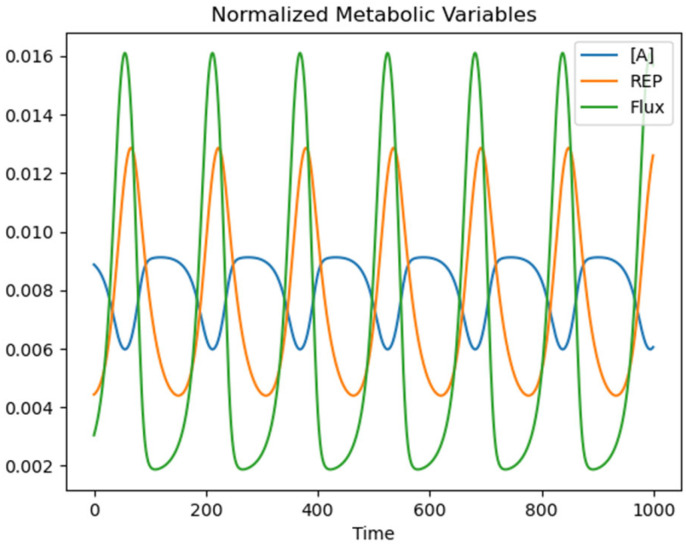
Time series of the internal concentration of A, REP, and Flux $J_{A}$. All values are normalized to appear on the same scale.

**Figure 10 entropy-24-01793-f010:**
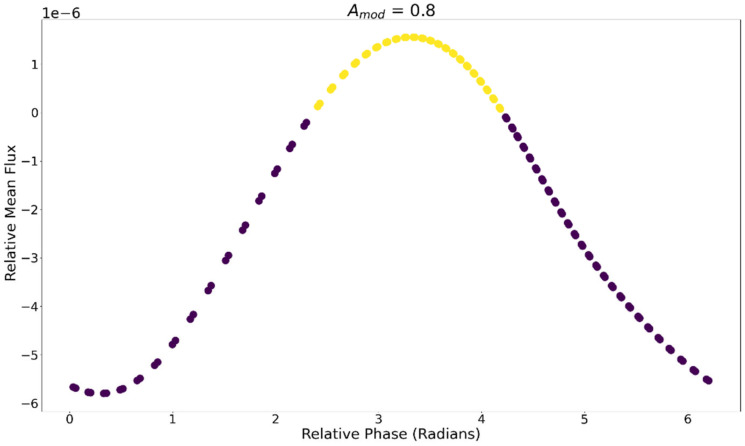
Average relative flux ∆J for different phase relations between motion and metabolism for $A_{mod}$ = 0.8. Positive values (yellow) indicate that the oscillatory dynamic had higher average $J_{A}$ values.

**Figure 11 entropy-24-01793-f011:**
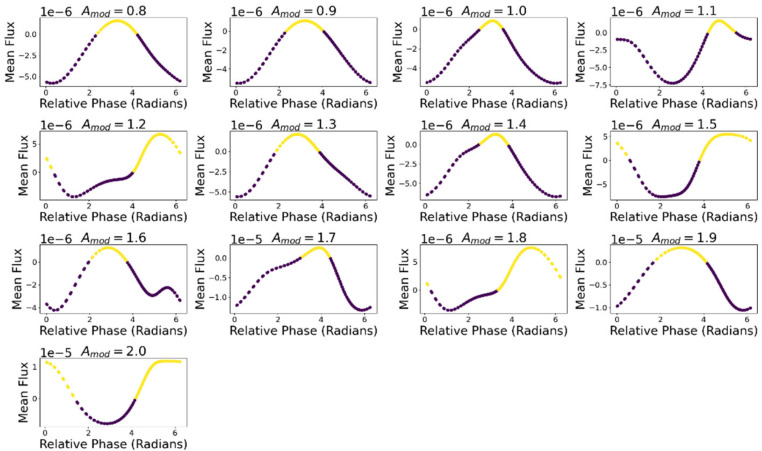
Mean relative flux ∆J for different phase relations between motion and metabolism, for each level of $A_{mod}$. Positive values (yellow) indicate that the oscillatory dynamic had higher average $J_{A}$.

**Figure 12 entropy-24-01793-f012:**
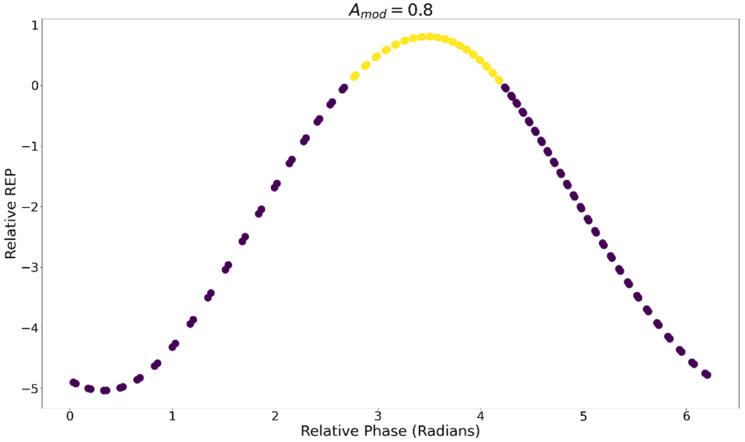
∆σ for different phase relations of motion and metabolism for $A_{mod}$ = 0.8. Yellow markers indicate that ∆σ is positive, and thus that the oscillatory dynamic produces more entropy.

**Figure 13 entropy-24-01793-f013:**
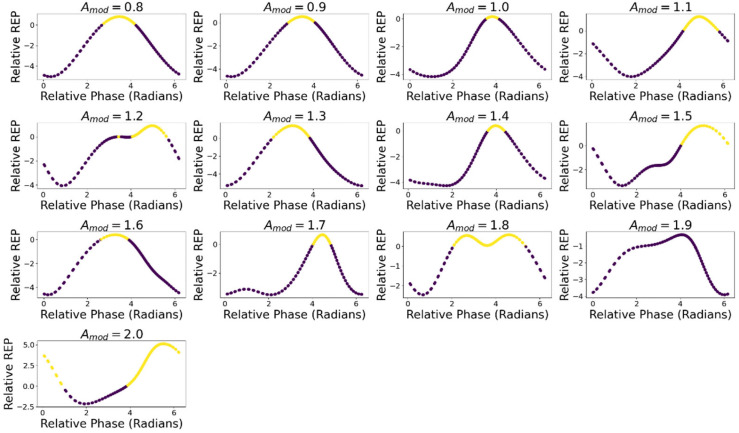
∆σ for different phase relations of motion and metabolism within levels of $A_{mod}$. Yellow markers indicate that ∆σ is positive, and thus that the oscillatory dynamic produces more entropy.

**Figure 14 entropy-24-01793-f014:**
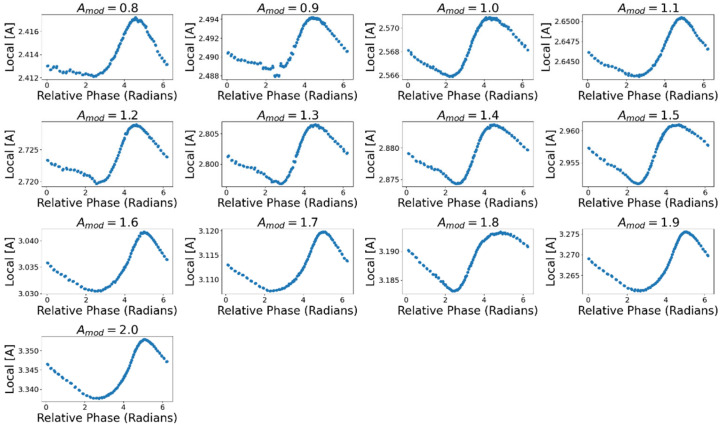
Average external concentration of A local to the droplet for different relative phase values. The local concentration peaks when relative phase is greater than π which corresponds with the droplet moving towards the midpoint of the distribution.

**Figure 15 entropy-24-01793-f015:**
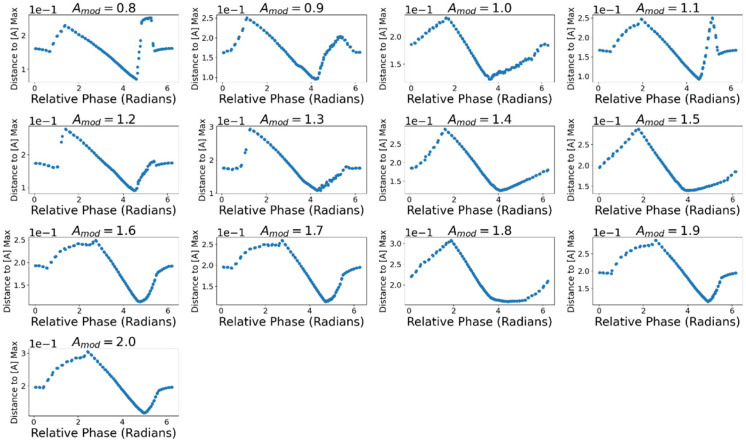
Average displacement of droplet from the point of maximum [*A*]. Displacement is minimized when $J_{A}$ peaks in the second half of the motive cycle.

**Table 1 entropy-24-01793-t001:** BZR Reaction Network.

	Reaction	Rates
R1	A+Y ↔X+P	K1
R2	A+X ↔2X+2Z	K2
R3	X+Y ↔2P	K3
R4	2X ↔A+P	K4
R5	Z ↔fY	K5
R6	P↔Y	K6
R7	Y+P ↔U	K7
R8	U+S ↔Q	K8

**Table 2 entropy-24-01793-t002:** Compounds and analogous metabolic functions.

Compound	Function
A	Fuel
X, Y	Initiators
U	Metabolic intermediate
Q	Waste

**Table 3 entropy-24-01793-t003:** Mass-action kinetics for all species within the cell. S is assumed to be present in excess on the surface of the cell and does not vary. $J_{A}$ is the rate at which *A* diffuses into the cell from the environment, parameter $J_{Q}$ represents the rate at which *Q* diffuses from the cell into the environment. *f* is a stoichiometric constant, not a chemical compound. Forward and reverse rate constants of reaction x are notated as $k_{fx}$ and $k_{rx}$ respectively.

$\frac{d\left[X\right]}{dt}=$	$\;k_{f1}\left[Y\right]\left[A\right]+k_{f2}\left[A\right]\left[X\right]-k_{f3}\left[X\right]\left[Y\right]-2k_{f4}{\left[X\right]}^{2}-$	(2)
	$k_{r1}\left[X\right]\left[P\right]-2k_{r2}{\left[X\right]}^{2}{\left[Z\right]}^{2}+2k_{r3}{\left[P\right]}^{2}+2k_{r4}\left[A\right]\left[P\right]$	
$\frac{d\left[Y\right]}{dt}=$	$-k_{f1}\left[A\right]\left[Y\right]-k_{f3}\left[X\right]\left[Y\right]+fk_{f5}\left[Z\right]+k_{f6}\left[P\right]-k_{f7}\left[Y\right]\left[P\right]+$	(3)
	$k_{r1}\left[X\right]\left[P\right]+k_{r3}{\left[P\right]}^{2}-k_{r5}{\left[Y\right]}^{f}-k_{r6}\left[Y\right]+k_{r7}\left[U\right]$	
$\frac{d\left[Z\right]}{dt}=$	$2k_{f2}\left[X\right]\left[A\right]-k_{f5}\left[Z\right]-$	(4)
	$2k_{r2}{\left[X\right]}^{2}{\left[Z\right]}^{2}+k_{r5}{\left[Y\right]}^{f}$	
$\frac{d\left[A\right]}{dt}=$	$-k_{f1}\left[A\right]\left[Y\right]-k_{f2}\left[A\right]\left[X\right]+k_{f4}{\left[X\right]}^{2}+J_{A}+$	(5)
	$k_{r1}\left[X\right]\left[P\right]+k_{r2}{\left[X\right]}^{2}{\left[Z\right]}^{2}-k_{r4}\left[A\right]\left[P\right]$	
$\frac{d\left[P\right]}{dt}=$	$k_{f1}\left[A\right]\left[Y\right]+2k_{f3}\left[X\right]\left[Y\right]+k_{f4}{\left[X\right]}^{2}-k_{f6}\left[P\right]-k_{f7}\left[Y\right]\left[P\right]-$	(6)
	$k_{r1}\left[X\right]\left[P\right]-2k_{r2}{\left[P\right]}^{2}-k_{r3}\left[A\right]\left[P\right]+k_{r6}\left[Y\right]+k_{r7}\left[U\right]$	
$\frac{d\left[U\right]}{dt}=$	$k_{f7}\left[Y\right]\left[P\right]-k_{f8}\left[U\right]\left[S\right]-$	(7)
	$k_{r7}\left[U\right]+k_{r8}\left[Q\right]$	
$\frac{d\left[Q\right]}{dt}=$	$k_{f8}\left[U\right]\left[S\right]-k_{r8}\left[Q\right]-J_{Q}$	(8)
$\frac{d\left[S\right]}{dt}=$	0	(9)

**Table 4 entropy-24-01793-t004:** Parameters for the motive oscillations. Amplitude and frequency were constant within levels of $A_{mod}$ while phase varied. Oscillations did not occur for $A_{mod}$ < 8 and are thus excluded.

Oscillation Amplitude	Oscillation Frequency	*A_mod_*
0.118408	4.030303	0.8
0.122086	4.13604	0.9
0.129831	4.240538	1.0
0.133508	4.393977	1.1
0.141784	4.497884	1.2
0.145548	4.607405	1.3
0.152022	4.712159	1.4
0.156634	4.817607	1.5
0.160864	4.965468	1.6
0.166069	5.071642	1.7
0.168857	5.131751	1.8
0.171134	5.281087	1.9
0.172585	5.386069	2.0

**Table 5 entropy-24-01793-t005:** Highest mean value of ∆J across different relative phase values for each value of $A_{mod}$, and the relative phase value $\Phi_{R}$ at which the maximum occurs.

*A_mod_*	Max ∆*J*	Φ*_R_*
0.8	1.56 × 10^−6^	3.34
0.9	1.10 × 10^−6^	3.18
1.0	8.77 × 10^−7^	3.10
1.1	1.67 × 10^−6^	4.78
1.2	6.72 × 10^−6^	5.28
1.3	2.11 × 10^−6^	2.83
1.4	1.33 × 10^−6^	3.26
1.5	5.45 × 10^−6^	5.07
1.6	1.26 × 10^−6^	2.88
1.7	2.61 × 10^−6^	3.91
1.8	7.53 × 10^−6^	4.84
1.9	3.16 × 10^−6^	2.97
2.0	1.18 × 10^−5^	5.57

**Table 6 entropy-24-01793-t006:** Displays the maximum ∆σ values and the ∆Φ values at which they occur.

*A_mod_*	Max ∆*σ*	∆Φ (Radians)
0.8	0.81	3.50
0.9	0.53	3.46
1.0	0.15	3.84
1.1	1.24	4.91
1.2	0.93	4.95
1.3	1.39	3.07
1.4	0.40	4.01
1.5	1.66	5.05
1.6	0.40	3.24
1.7	0.66	4.48
1.8	0.58	4.61
1.9	−0.33	4.06
2.0	5.12	5.51

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
