# Peer review of "Foraging Dynamics and Entropy Production in a Simulated Proto-Cell"

_entropy, 2022, doi:10.3390/e24121793_

Round 1

Reviewer 1 Report (Previous Reviewer 3)

My main concern is the magnitude of the REP difference, which now is somehow addressed by the statistical significance test. I think it’s fine.   I do not have any other concerns now.

Author Response

Thank you for your time reviewing the revised manuscript.

Reviewer 2 Report (Previous Reviewer 2)

No significant changes have been made in the resubmitted work. There are only cosmetic corrections. Based on the authors' responses, it is clear that they have only just begun to deal with nonequilibrium thermodynamics and entropy production. If the authors would completely exclude any mention of entropy production and principles of minimum/maximum, and include only simulation of chemical dissipative structure in their paper, the paper could be considered for publication. If, of course, authors believe that in this case the paper will have novelty.

Author Response

Thank you for reviewing the revised manuscript. We would like to reiterate that there is a common and respectable practice of using simulated chemical systems to generate claims about entropy production and related variational principles. In incomprehensive list of some such works is included below. For this reason, we have not removed the discussion of entropy production as suggested.

Irvin, B. R.; Ross, J. Calculation of the Rate of Entropy Production for a Model Chemical Reaction. J. Chem. Phys. 1988, 89 (2), 1064–1066. https://doi.org/10.1063/1.455258.

Månsson, B. G. Entropy Production in Oscillating Chemical Systems. Zeitschrift fur Naturforsch. - Sect. A J. Phys. Sci. 1985, 40 (9), 877–884. https://doi.org/10.1515/zna-1985-0903.

Kachman, T.; Owen, J. A.; England, J. L. Self-Organized Resonance during Search of a Diverse Chemical Space. Phys. Rev. Lett. 2017, 119 (3), 1–5. https://doi.org/10.1103/PhysRevLett.119.038001.

Kondepudi, D. K.; Kapcha, L. Entropy Production in Chiral Symmetry Breaking Transitions. Chirality 2008, 20, 424–528.

Kondepudi, D.; Mundy, Z. Spontaneous Chiral Symmetry Breaking and Entropy Production in a Closed System. Symmetry (Basel). 2020, 12 (5). https://doi.org/10.3390/SYM12050769.

Endres, R. G. Entropy Production Selects Nonequilibrium States in Multistable Systems. Sci. Rep. 2017, 7 (1), 1–13. https://doi.org/10.1038/s41598-017-14485-8.

Reviewer 3 Report (Previous Reviewer 1)

The authors have made significant improvement on the paper. I recommend publication.

I still find a few typos or inconsistencies that should be corrected:

- Fig 1 caption: "propogotating";

- Line 284: both [A]_i and [A_i] appear; Line 291: [A_i], Line 292: [A];

- "Acknowledgments" is inconsistent with "Author Contributions", since one of the authors is thanking another author;

Author Response

Thank you for your attention to detail in the review process. 

  • The spelling error has been remedied
  • The notation has been made consistent to distinguish between the concentration of A at each location in the distribution ([Ai]) and the concentration of internal to the droplet ([A]).
  • The acknowledgement of Dr. Kondepudi has been removed.

Round 2

Reviewer 2 Report (Previous Reviewer 2)

The authors disagree with my suggestion to improve the manuscript. They insist on publishing an invented hypothetical mathematical model, which is in no way verified by experiment and calls into question the basic concepts of physics. In my opinion such state of affairs is absurd for a physics journal. It is inadmissible for the reputation of the journal to publish such a paper.

This manuscript is a resubmission of an earlier submission. The following is a list of the peer review reports and author responses from that submission.

Round 1

Reviewer 1 Report

The authors propose a mathematical model for a minimalistic protocell where the internal reaction network is coupled to the external movement of the droplet as a whole. The model refines the properties of a hypothetical food-like ingredient A with respect to similar models presented in the literature by including its variations in time and space. The authors study whether the Maximum Entropy Production Principle (MEPP) underlies the nonequilibrium dissipative dynamics of the protocell, finding either positive or negative results, depending on the external parameters. They also investigate how the details of the dynamics, specially synchronization, could explain their results.

The subject is thought-provoking, addressing questions at the boundaries between living and nonliving systems. The model is clearly justified, as it is grounded on experimental work. Results are convincing, and well explained. The paper is well written, with pictures and tables nicely presented.

Before recommending acceptance of the paper, there are some points I believe should be improved in the final version of the manuscript.

1) A central theme of the paper is entropy production. The authors make considerable effort to make sense of the violations of the MEPP they find in their model. It seems to me that there are close analogies between the scenarios studied in this paper and the models discussed in (i) T. Kachman, J. Owen, and J. England, Phys Rev Lett 119, 038001 (2017), and (ii) H. Kedia et al, arXiv:1908.09332. In both (i) and (ii), we see physically meaningful examples showing how a system may sometimes stabilize at maximum entropy production (energy-seeking) or at minimum entropy production (energy-avoiding) states. In (ii), this transition is smooth and quite emblematic: bistable springs absorb a lot of energy from a near-resonance drive of increasing amplitude, until it reaches the other stable far-from-resonance minimum, turning out to be of low absorption. What motivates me to quote these two papers is that a candidate for unifying principle is put forth, namely ‘dissipative adaptation’, encompassing those two apparently contradictory behaviours (max and min entropy production).

My question is: Wouldn't it be the case that the counterexamples of the MEPP discussed in the present paper (i.e., the cases of negative \Delta \sigma) could be regarded as somehow analogous to the bistable systems described in (i) and (ii)? I think that a comment on that point would considerably enhance the quality of the paper.

2) Below I list some typos that I think should be corrected, as well as some notations that I believe should be clarified:

2.1- what is a chemical wave (see caption of Fig.1)? Do the authors mean chemical oscillation?

2.2- BZ is first used in line 127, but first defined in line 133;

2.3- section 1.3 is misnumbered; 1.2 instead?

2.4- line 243: “of the concentration of the internal concentration of”;

2.5- Eq.14: How do the authors deal with the divergence at “xi = xc”?

2.6- Eq. 14: the definition of [Ac] is not clear to me. Is it true that [A] = [Ac] in Eq.(5)? Please explain.

2.7- Eq.16 and Eq.5: If [A] is the concentration inside the cell, then shouldn’t J_A be replaced by J_A(x_c)? Again, definitions are unclear to me. Also, is the velocity “v” the derivative of “xc”?

2.8- Line 603: “suggesting that that the droplet”.

Reviewer 2 Report

The work makes an extremely depressing impression. The article is devoted to modeling a hypothetical process, which is impossible to compare with an experiment. Of course, the authors do not do this. Accepting a lot of assumptions and approximations, taking values of coefficients from somewhere - they do not ask the question - is it possible in nature? If the purpose of the work is a mathematical exercise - then, what novelty in mathematics? That novelty, of course, does not exist either. To somehow justify the purpose of their work, the authors refer to MEPP and say they want to test it. But, MEPP is a law of nature and it is not very clear why some invented arbitrary mathematical model should satisfy it. If, for example, the authors will see that their model did not satisfy the law of conservation, they would adjust the model. Why, then, in the case of MEPP, do they test the principle with the model, instead of doing the opposite! Thus, the work is methodologically meaningless.

More specific remarks: 1) The authors refer to Ref.10, but apparently have not read it. As a consequence, the authors "test" MEPP outside the applicability of this principle.

2) Why introduce a new term: REP? It introduces confusion. Why introduce the second derivative of entropy, MEPP uses the first derivative of entropy?

The work cannot be published.

Reviewer 3 Report

The authors studied the dynamics and thermodynamics (entropy production) in a simulated proto-cell system. The nonlinear system shows transition between a single steady state and limit cycle, with coupling to different foraging behaviors. The motivation of using physical principles like the maximum entropy production principle to understand living system, is profound. However, the results shown in this study is very preliminary and might need more systematic exploration for a concrete conclusion. For general purposes, I have two suggestions:

1.    The difference of the entropy production between the two modes is negligible (Fig. 6,7&8), less than 0.5%. As for the core results of this manuscript, I wound say there is little difference, given the current parameters sets. 

2.    In non-equilibrium systems, the thermodynamics often depends on the dynamics of the system. There is no reason that a limit cycle would cost more than a static steady state, or vise versa. However, to connect physics and living system, a natural and interesting question in this foraging system could be, how the foraging behavior would be related to the entropy production. For example, is it true that a better forager would cost more?